# Identification and classification of urban employment centers based on big data: A case study of Beijing

Liang Wang[1,2]*, He Cui[2]

1 School of Economics and Management, Beijing Jiaotong University, Beijing, China, 2 Beijing Municipal Institute of City Planning & Design, Beijing, China

* wljean@126.com

**Data Availability Statement:** The use of the dataset was done in compliance to the BaiduHUIYAN and State Administration of Market Supervision and Administration(SAMSA) Privacy Policy (https://map.baidu.com/zt/client/privacy/

## Abstract

The layout, scale and spatial form of urban employment centers are important guidelines for the rational layout of public service facilities such as urban transportation, medical care, and education. In this paper, we use Internet cell phone positioning data to identify the workplace and residence of users in the Beijing city area and obtain commuting data of the employed to measure the employment center system in Beijing. Firstly, the employment density distribution is generated using the data of the working places of the employed persons, and the employment centers are identified based on the employment density of Beijing. Then, we use the business registration data of employment centers to measure the industrial diversity within the employment centers by using the ecological Shannon Wiener Diversity Index, and combine the commuting links between employment centers and places of residence to measure the energy level of each employment center, analyze the hinterland and sphere of influence of each center, and finally using the industrial diversity index of employment centers and the average commuting time of employed persons, combined with the K-Means clustering algorithm, to classify the employment centers in Beijing. The employment center identification and classification method based on big data constructed in this study can help solve the limitations of the previous employment center system research in terms of center identification and commuting linkage measurement due to large spatial units and lack of commuting data to a certain extent. The study can provide reference for the regular understanding and technical analysis of employment centers and provide help for the employment multi-center system in Beijing in terms of quantifying the employment spatial structure, guiding the construction of multi-center system, and adjusting the land use rules.

## 1. Introduction

The layout, scale and spatial morphology of employment centers in cities have important guiding significance for the rational layout of public service facilities such as urban transportation, medical care and education [1, 2]. In recent years, with the increasing scarcity of land resources, the optimization of urban space and the improvement of urban functions have become particularly important in the era of stock planning. The location, scale and functional classification of employment centers have naturally become the focus of attention of relevant

index.html, https://bj.gsxt.gov.cn/affiche-query-info-help-110000.html), which stated that the anonymized and aggregated data could be used for other services. This study was carried out with the support of BaiduHUIYAN and SAMSA, and the use of anonymized and aggregated data is authorized by BaiduHUIYAN and SAMSA and its Terms and Conditions. However, some restrictions will apply when accessing the original data, and the data request should be addressed to BaiduHUIYAN (https://huiyan.baidu.com/products/platform, E-mail: huiyan@baidu.com) and State Administration of Market Supervision and Administration (https://bj.gsxt.gov.cn/index.html, Tel: 86-010-82691819).

**Funding:** This research is supported by the National Key R&D Program of China (Nos. 2021YFA1000300 and 2021YFA1000304). The funders had no role in study design, data collection and analysis, decision to publish, or preparation of the manuscript.

**Competing interests:** The authors have declared that no competing interests exist.

managers and researchers [3–6]. The uneven distribution of urban public service facilities, traffic congestion caused by the separation of jobs and housing, and other big city problems have also prompted city managers and policy makers to reconsider the urban pattern [7], and the identification of employment center boundaries has gradually become the focus of scholars [8], and its location, scale and functional classification have naturally become the focus of relevant managers and researchers.

At present, many scholars have carried out related research work, generally using the population density index of employment centers to determine the absolute and relative criteria, the traditional employment center boundary determination mainly relies on the combination of field research and city-related economic statistics analysis method [7]; the fitting models of employment density are non-parametric models [2, 7] and parametric models [9], and the Clark model is found to be more effective in fitting the change of population density in the pattern of urban centers. while the Smeed model is superior in fitting the citywide geographic distribution of population including the suburbs [10]; while the non-parametric models can achieve better results in fitting the change in employment population density in the polycentric city structure [9]. Currently, urban employment centers are developing in the direction of polycentricity, systematization and networking, etc., limited by the fact that the spatial units of data mainly used in current studies may be too large (e.g., the smallest unit of data such as the economic census is the street) and cannot be subdivided within the larger spatial units, resulting in the identified employment center boundaries only overlapping with one or several spatial units, which may have large deviations from the actual boundaries [3, 11, 12], and the lack of research on measures such as the commuting direction and range of employed persons in employment centers and the types of employment centers, limited by survey means and the lack of basic information are the main reasons for the above limitations. With the popularity of mobile communication in recent years, the spatio-temporal trajectories of mobile users can be recorded by base stations, which greatly improves the spatial accuracy of the data. For example, the spatial accuracy of cell phone positioning data obtained by Internet companies can reach 100-meter grid (1 hectare), which is much smaller than the average area streets in Beijing of 4841 hectares.

In this paper, we use cell phone positioning data and enterprise business registration data in Beijing to identify the spatial location, scale, and functional classification of employment centers in Beijing based on the distribution of individual employed people's workplaces and the commuting links between workplaces and residences to provide basic information for urban research, planning preparation and urban management. This paper attempts to focus on the following four aspects on a smaller spatial unit: (i) employment center identification: how jobs are distributed and which areas are most densely employed as a way to identify employment centers; (ii) employment center industrial diversity analysis: using the Shannon-Wiener diversity index, combined with data on business registration of enterprises in employment centers, to calculate the industrial diversity of employment centers; (iii) employment center attraction and radiation range: analyze the radiation range of employment centers by using the commuting links of the employed in the employment centers; (iv) classification of employment centers: Industrial diversity index of employment centers and average commuting time of employed persons, combined with K-Means clustering algorithm, for classification of employment centers in Beijing.

## 2. Literature review

### 2.1 A review of employment center system measurement methods

The development of modern urban spatial structure theory can be traced back to the 1960s, when three urban economists, Alonso, Muth, and Mills [13–15] proposed the monocentric

city model. In the following four decades, this abstract theoretical model has been used and can provide a reasonable explanation of urban. Due to the combined effect of urbanization rate, building height, building density and housing consumption, the urban population in monocentric cities shows a regular spatial distribution—the population density decreases with increasing distance from the city's business center (CBD). In the monocentric city model, employment is highly concentrated, so the population density and its rate of change (i.e., gradient) can more adequately reflect the spatial structure characteristics of the city, which makes urban population density the most concerned variable when studying urban spatial structure. Since the study of the Clark model was proposed in 1951 [16], many scholars have worked on models that estimate different urban population density functions and their variation patterns with space [17]. The monocentric city model explains the location and travel behavior of city residents given the exogenous location of firms and the resulting characteristics of the urban spatial structure. Enterprises in actual cities choose their locations considering urban land cost, production cost, labor availability and market proximity. Along with the rapid development of cities, the congestion effect of urban centers gradually emerges and the production cost of enterprises gradually increases, which also motivates enterprises to gradually look for space at the periphery of cities, which in turn leads to the splitting of employment centers and the formation of polycentric urban forms, and more and more sub-centers appear within cities, so that the spatial distribution of land (housing) prices, population, and employment in cities will change [3, 18], but the urban The dominant role of the main center is still stronger, forming a primary and secondary polycentric pattern [19].

In terms of employment center identification, some international scholars have given a general employment center identification method, which generally subdivides cities into multiple regions and selects employment centers by analyzing the changes in the number of jobs in adjacent regions [18, 20, 21]. Giuliano and Small et al [21] proposed that employment centers are defined as areas with employment densities greater than 2,500 persons per square kilometer and employment numbers greater than 10,000 persons, and identified 32 employment centers in Los Angeles based on the Los Angeles transportation survey data among 1146 transportation subdivisions (average area of each transportation subdivision is about 800 ha). The 32 employment centers in Los Angeles were identified, but due to the large area of the traffic cells used, employment centers with smaller areas or less density could not be accurately identified, and the boundaries of the identified employment centers were usually only the boundaries of the traffic cells [19]. McMillen and Daniel [7] used traffic survey data from cities such as Chicago, and proposed local weighted regression based on the assumption of monocentric spatial structure and semi-parametric regression method, which solved the problem of identifying lower density employment centers, but still could not identify employment centers with smaller area due to the limitation of statistical cell area (about 1000 ha). Vasanen [22] used 250m * 250m raster commuting data to identify employment centers in three Finnish cities with simpler employment density local spatial autocorrelation (Local Moran's I), avoiding the effect of statistical cell size and identified the cluster with the largest area as the main center and the remaining clusters as secondary centers. However, its required data accuracy is high and it is difficult to obtain. In summary, urban center identification and circle delineation scholars identify urban centers mainly based on employment density or population density data, using employment or population density peak, threshold method, parametric method and semi-parametric method to identify urban centers, in the research method generally use data distribution characteristics to outline the hot spot areas of the research object, including urban employment and residential hot spots, urban crime hot spots, tourism hot spots, etc [8, 23]. In this paper Employment center identification mainly combines the methods proposed by Giuliano or Mcmillen [12, 24].

In terms of employment center classification, employment centers, as the locations where commercial activities are concentrated in cities, are responsible for residents' business and leisure functions, and are the core area of urban economic and social activities. The American scholar Proudfoot [25] classified American urban employment centers into five types according to the spatial location and characteristics of business activities: central employment centers, peripheral employment centers, major employment centers, neighboring employment centers, and clusters of isolated business sites. Davies et al. [26] conducted an empirical study of British towns and cities and found that the specialized business territories in Britain were not as obvious as in the United States. Lamb [27] studied the shape and dynamics of business districts in 40 small towns in upstate New York. In China, research on the types and spatial structure of business territories has been gradually carried out since the 1990s [28]. Chen et al. [29] used to classify the locational types and service levels of retail business services in the downtown area of Kunming, and Wu et al. [30] used GIS spatial analysis to delineate the hierarchy of business centers in Beijing by constructing an index system. Zhang et al. [31] used GIS point pattern analysis to compare and study the distribution and spatial clustering characteristics of business districts in Beijing in 2004 and 2008. From the above analysis, the previous studies on the geographical structure of urban business started earlier and a series of research results were formed. However, in general, due to the data, the spatial scale of previous studies on the spatial scope definition and classification of intra-city business districts is too large or differs greatly from the actual situation. In recent years, some scholars have obtained basic data through a large amount of research in order to study the business district, which is a large amount of work to obtain data and leads to a large research cost [32], while the quantitative identification as well as classification of urban business districts is of great value to the study of urban business development and the study of urban planning and construction.

## 2.2 Employment center research exploration with big data

Currently, the main studies on employment center identification and classification require data on employment jobs, enterprise data, and commuting. At present, although the economic census in China can obtain the employment job data of each enterprise, the smallest spatial unit of publicly available data is the street, which may be too large relative to the employment center or not coincide with the actual employment center boundary [24], which may reduce the actual population density of the employment center, resulting in the identification results not matching the actual. Although the Beijing traffic survey data can simultaneously obtain commuting data of employment centers, the sampling rate of the traffic survey is low and the spatial unit precision of the aggregated public data is like that of the economic census, which cannot substantially help the study. Big data (such as cell phone positioning data, cell phone call data, cab GPS data, bus IC smart card data, etc.) that have emerged in recent years record a large number of users' spatio-temporal trajectories, from which data analysis can be used to obtain data such as human flow density or commuting connections, and scholars have used these data to conduct urban spatial studies [33–35], and the spatial unit can generally be controlled at about 100 ha, and the statistical The number of users is often above the million level, which to a certain extent helps to address the impact caused by past data deficiencies.

The cell phone location data utilized in this paper records in real time the GPS location of cell phone users' cell phone connection when they use Internet location services, from which the information of users' occupational and residence, commuting, etc. can be obtained by analyzing the spatio-temporal trajectories of cell phone users as the basic data for employment center identification research, and the business registration information of enterprises in employment centers is used to analyze the diversity of enterprises in employment centers with

the help of Shannon-Wiener diversity index, combined with commuting data of employed persons in employment centers to classify employment centers. Currently, scholars have used cell phone data to study issues related to employment commuting through workplace and residence identification, for example, Ahas et al. [36] used cell phone call data for 12 consecutive months to identify about 45% of users' workplace and residence; Becker et al. used cell phone call data recorded at 35 base stations within 13 square kilometers in Morristown, New Jersey, over two months to identify users' workplace and residence, and used the identification data to analyze where employed people live in Morristown to obtain the commuting range of the city. Using cell phone location data, Kan et al. [37] identified the workplace and residence of employed people in the city and analyzed their commuting characteristics, and verified the reliability of the data. The above-mentioned residential population distribution identified by cell phone data can reflect a more realistic population distribution, but existing studies have not explored the identification and classification of employment centers using big data.

## 3. Materials and methodology

### 3.1 Study area

In this study, Beijing is selected as a case study. There are 16 districts in Beijing, with a total area of 16,400 square kilometers. Among them, Dongcheng and Xicheng are the Core District; the central core district and six districts, namely Chaoyang, Haidian, Fengtai and Shijingshan, are collectively referred to as the Central District; Tongzhou, Changping, Shunyi, Daxing, Mentougou and Fangshan become the Suburban District; and four districts, namely Huairou, Pinggu, Miyun and Yanqing are four suburban areas, as shown in Fig 1. We chose this city for several reasons. First, as the capital of China, it has been a model for Chinese cities in terms of land and development market development, urban planning and policy making. As such, Beijing is well established as a field of study for the economic, social, and spatial transformation of

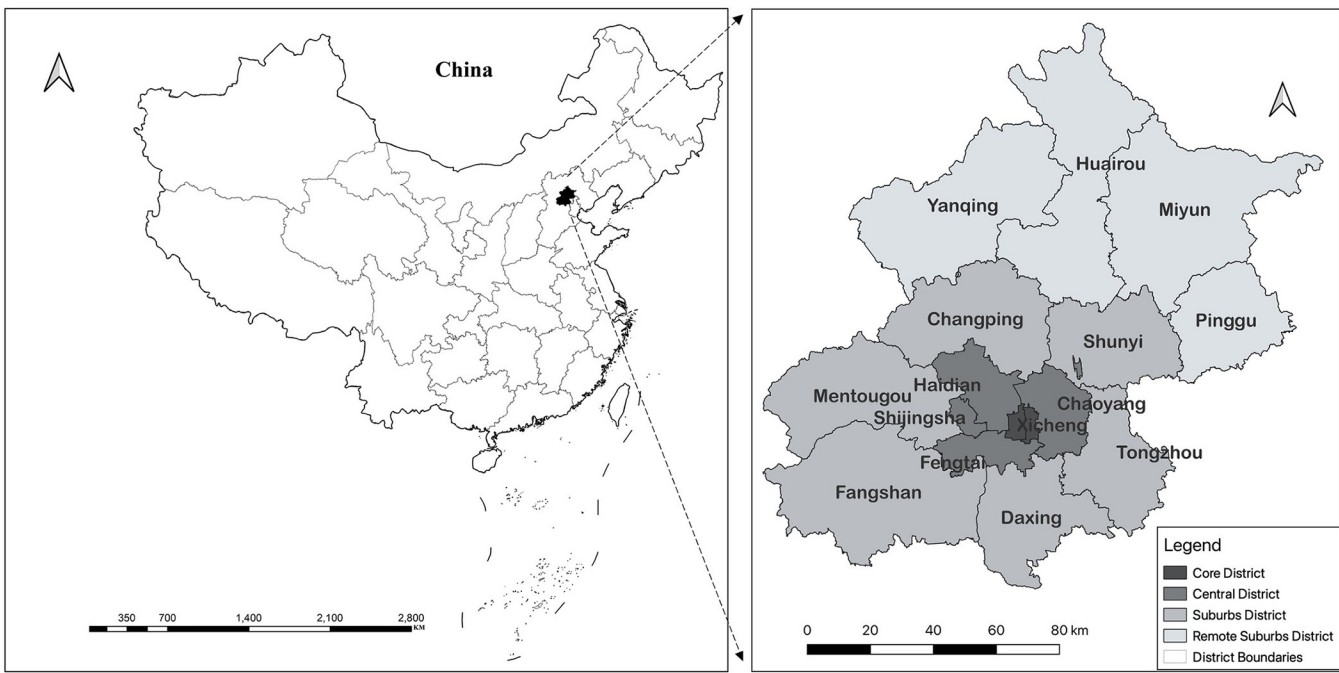

**Fig 1. Research area of the study.** Reprinted from [https://www.resdc.cn/, accessed on 10 May 2021], original copyright 2021.

Chinese cities. Second, Beijing is growing rapidly, and given this high rate of development, it is important to study short-term changes in Beijing. At the same time, short-term changes can be used to assist in long-term evolutionary studies. Again, with the rapid expansion of Beijing, several urban employment centers have emerged in the city, and the number, size, attractiveness, and evolution of these employment sub-centers merit further study. Finally, in order to alleviate urban problems, the Beijing government has implemented a series of programs and policies to promote decentralized development and attempt to form employment sub-centers in the peripheral areas of Beijing. The construction of the subway is an important means of implementing these policies, and the effectiveness of this approach needs to be tested.

## 3.2 Data

In this paper, we use basic spatio-temporal big data such as Internet location services data (also known as cell phone positioning data), business enterprise registration data, etc. Each type of data is anonymized in all aspects of processing, and no individual privacy is involved in all aspects and outputs.

**3.2.1 Internet location service data.** The Internet location service data (also known as cell phone location data) used in this paper is obtained based on Baidu Map location service requests, and the source of the data is mainly from Baidu Map users agreeing to the corresponding terms and conditions of service of the Internet service provider through an app such as Baidu Map. The user data collected without personal information using the connection between the communication base station network and the user's mobile device, which mainly includes both temporal and spatial dimensions, is used to provide services such as navigation for the user's movement within the city. At present, the average daily location service requests of Baidu Maps exceed 120 billion, and the number of active smart devices exceeds 1.2 billion per month, covering all levels of administrative divisions in China. Location service data is the basis of urban occupancy, commuting and other features mining, mainly used for occupancy space analysis, commuting OD and commuting time analysis. Location data assists in extracting samples on the one hand, including the extraction of resident points (residence, employment, etc.), and the extraction of samples of commuting modes such as cars, railways, and buses; on the other hand, it is used to construct feature data, including the construction of features such as the number of locations in resident points and the distribution of location time [38].

**3.2.2 Enterprise business registration data.** The enterprise data used in this paper is from the business registration data of the industry and commerce department, which contains the information of all enterprises in Beijing from October 1949 to May 2021, including the unique identification number, company name, industry code, address, industry, registered capital and registration status (surviving, cancelled and revoked, etc.), with a total data volume of 4.21 million. The fields used in the study are listed in Table 1. The industry code field is used to determine the industry category of the enterprise; the address and region fields are

Table 1. Business registration data fields used and their purpose.

| Field Name | Example | Applications |
|---|---|---|
| Industry Code | IRS | Extract the enterprise industry |
| Address, area | No. 10, Shangdi 10th Street, Haidian District, Beijing | Extract the workspace location |
| Industry | Internet and related services | Determine the type of business |
| Registered Capital | 45.2 million USD | Extract the size of the enterprise |
| Registration Status | Survival | Extract enterprise status |

used to determine the spatial location of the enterprise, and the longitude and latitude coordinates of the enterprise are obtained by inverse address resolution using Baidu Map API; the registered capital field is used to determine the scale of the enterprise; and the enterprise registration status is used to determine the status of the enterprise. Due to the limitation of space, the technical details of data pre-processing will not be discussed in this paper [39].

## 3.3 Methodology

**3.3.1 Ethics statement.** This research was granted ethical approval from the Ethics Committee at Beijing Municipal Institute of City Planning and Design, China. The consent was not needed for this study because the data used in the study was anonymized by baidu before being transferred to the authors, in such a way that no connection could be made with any individual. The mobile phone users agreed to the terms and conditions for using the services with the internet service provider that their communications may be recorded and analyzed for improving services.

**3.3.2 Employment center identification method.** In this paper, based on the cell phone location data in Beijing in May 2021, we use the spatial clustering DBSCAN (Density-Based Spatial Clustering of Applications with Noise) algorithm to cluster the location points and obtain multiple independent clusters; and extract location, land use attributes, user portraits, etc. from the clusters over 60 features, classify the user clusters using machine learning algorithms such as XGBoost (eXtreme Gradient Boosting), get the residence, employment, and recreation places of users [37], and finally identify the residence and employment places of the employed population in Beijing in May 2021, gather the employed population spatially on a 100 m * 100 m raster grid, and use kriging spatial interpolation, generating equipotential surfaces, and combining with local empirical knowledge, the employment center is defined as an area enclosed by equipotential surfaces, whose area is not less than 1 km2 and not more than 10 km2, and whose population density is greater than 10,000 people/km2, and the number of employed people in the area is not less than 20,000, so as to identify the employment center of Beijing, a mega metropolitan area.

DBSCAN is a density-based clustering algorithm, in which the number of objects (points or other spatial objects) contained in a certain region of the clustering space is not less than a given threshold, and the algorithm can discover arbitrarily shaped clusters from a noisy data set, so that regions with sufficient density are divided into the same cluster to achieve the purpose of clustering. The significant advantage of DBSCAN is that it can effectively eliminate noisy data and discover arbitrarily shaped spatial clusters efficiently and quickly. In this study, we mainly cluster the valid data on all users and obtain the arithmetic mean of coordinate points in the largest clusters so that each user can get at least two valid clustering points [40]. xGBoost is an integrated learning algorithm that constructs a strong classifier by training multiple weak classifiers for supervised learning classification of users' residence and place of employment, etc. By constructing a commuting mode mining classification model based on machine learning algorithms such as Bayesian, SVM, decision tree, random forest, GBDT and XGBoost, Kan et al. [37] trained the model using a sample set and 71 features, and compared, tested and evaluated different machine learning algorithms and found that the XGBoost algorithm had the best accuracy and recall rate of over 87% overall.

**3.3.3 Employment center industry diversity analysis.** The number, scale, type, and spatial distribution of industrial and commercial registration of enterprises in cities can reflect the concentration degree and difference of various industries in space. It is of positive significance to analyze the number, scale, and type of industrial and commercial registration of enterprises in various employment centers and analyze the advantageous areas of their distribution for the

recognition of urban employment centers. This research innovates the use of the Shannon Wiener Diversity Index, which describes biological diversity, to analyze the industrial diversity of employment centers. The Shannon Wiener Diversity Index was proposed by Claude Elwood Shannon, the founder of information theory, in 1948. It is widely used in mathematics, communications, ecology and other disciplines, mainly to measure the species richness within the system and the uniformity of individual distribution in various categories, It can also be applied in the field of economics to measure the industrial diversity of a job center, where industrial diversity is an important advantage because it provides a wider range of employment opportunities and mitigates the risk of a particular industry being hit during a recession. In the calculation process, the data are categorized by industry type, the percentage of each industry in total employment is calculated, and the Shannon Wiener Diversity Index formula to calculate the industrial diversity index of that job center, according to the index can assess the strengths and weaknesses of the industrial diversity of the job center, as well as the comparison with other job centers, and formulate policies accordingly. This research uses the enterprise information registered in the employment center (business registration data) to calculate the industrial diversity of the employment center. The calculation formula is as follows:

$$H\prime = -\sum_{i=1}^{S} P_i * \log_2 P_i \tag{1}$$

Where, $H'$ indicates the information diversity of the research object, and in this paper, it represents the industrial diversity index of the employment center, $S$ indicates the number of enterprise types, and $P_i$ represents the proportion of registered capital of enterprises in category $i$, and from the above expression, we can see that $H'$ obtains a great value when $P_i = 1/S$. The industrial diversity index $H'$ reflects the diversity and uniformity of each enterprise type in the employment center. Under the condition that the number of industrial types is approximately the same, the more uniform the development of industrial types, the larger the value of its diversity index; on the contrary, the more prominent the dominant industry and the more aggregated the industrial types, the smaller the value of its diversity index $H'$.

**3.3.4 Employment center classification method.** In this study, we use cell phone location data to analyze the residence of employed people in each employment center and use their average commuting time as the attraction and radiation range of each employment center, and use both this and the industrial diversity index of the employment center as the classification condition to classify the identified Beijing employment centers using K-Means clustering algorithm, K-means is a common clustering algorithm that divides data points into K different clusters, each of which has similar characteristics and has significant differences between data points in different clusters. Since K-means is an unsupervised learning algorithm that does not require a priori knowledge, it can automatically classify employment centers based on their industrial diversity and the average commuting time of employed people, the method has the following advantages such as high scalability, robustness and interpretability of results. The technical route of employment center identification and classification in this paper is shown in Fig 2.

## 4. Results

In this paper, we use the Baidu cell phone location data in May 2021, including the encrypted unique user identification number, the time when the location occurred, the GPS location of the phone when the location occurred and other information, and use machine learning algorithms such as XGBoost to identify 27.87 million people living in Beijing and 13.62 million people employed, and extract the commuting OD based on the place of residence and

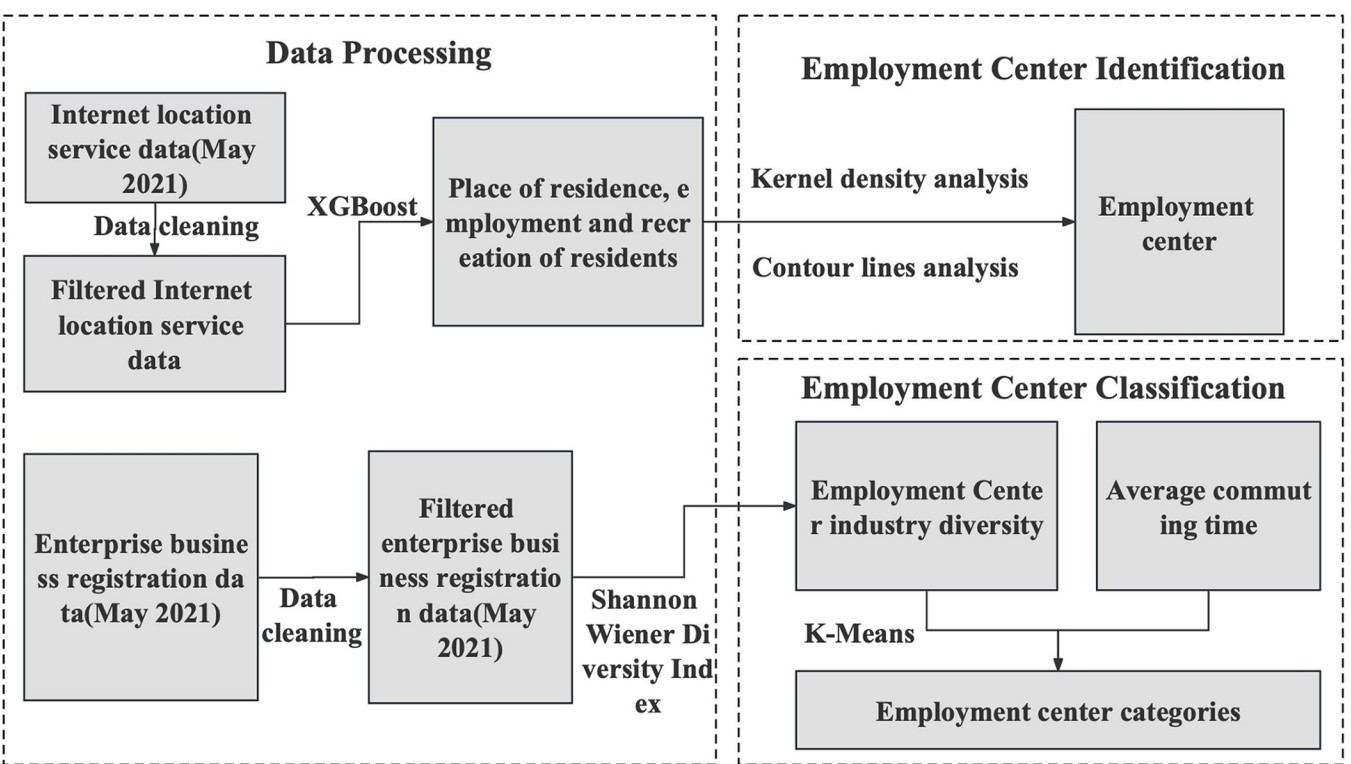

**Fig 2. Technology roadmap.**

employment to get the data on the place of work and employment and commuting time of the employed population, whose user commuting data are shown in Table 2.

The commuting data of the employed persons obtained from the cell phone location data are aggregated by the grid representing the workplace to obtain the number of employed persons in each grid, since in the reality employed persons are not exactly located at the location of the grid center point, but at a location within the grid coverage. To simulate the real density distribution of the employed persons, the kernel density (Kernel Density) analysis is done in QGis with 800m as the search radius, and the number of employed persons connected to each base station is apportioned into a 100m×100m grid, and the attribute value of each grid represents the employment density of that grid, and its employment density distribution is shown in Fig 3.

## 4.1 Employment results and verification

To test the accuracy of the above identification methods, this paper uses the linear correlation between the resident population data from the 7th census of China in 2020 and the entire

**Table 2. The attributes of residential employment and commuting data of mobile users.**

| User ID | Grid ID of workplace | Location of workplace | Grid ID of residence | Location of residence | Commuting time (hours) |
|---|---|---|---|---|---|
| X1 | 712XXXXX | 116.21XX,39.23XX | 722XXXXX | 116.26XX,39.29XX | 0.5XX |
| X2 | 713XXXXX | 116.28XX,39.92XX | 713XXXXX | 116.51XX,39.33XX | 0.8XX |
| X3 | 722XXXXX | 116.25XX,39.25XX | 735XXXXX | 116.71XX,39.56XX | 1.2XX |
| X4 | 732XXXXX | 116.81XX,39.53XX | 715XXXXX | 116.91XX,39.41XX | 0.9XX |
| X5 | 714XXXXX | 116.31XX,39.13XX | 729XXXXX | 116.92XX,39.83XX | 0.2XX |

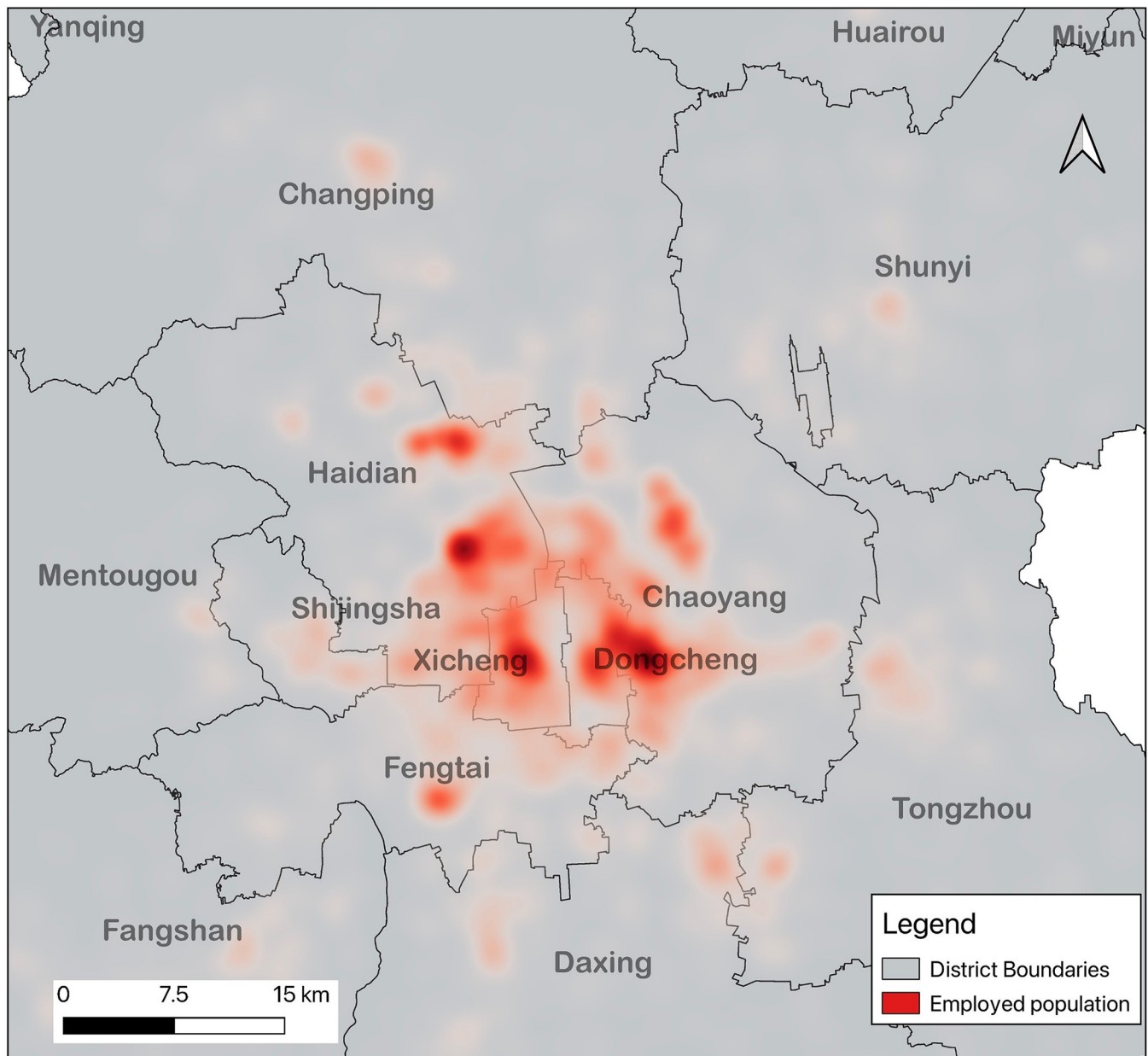

**Fig 3. Heat map of employment distribution in Beijing.** Reprinted from [https://www.resdc.cn/, accessed on 10 May 2021], original copyright 2021.

residential population identified with cell phone location data, this paper uses the linear correlation between the resident population in 339 streets in Beijing from the 7th census of China in 2020 and the entire residential population identified with cell phone location data, the 7th census data is China's seventh national census, conducted at 00:00 on November 1, 2020, mainly includes data on the number, structure, and distribution of China's population, and its spatial minimum scale is streets (or townships). The two were calculated to be positively correlated and passed a 99% confidence interval test, as shown in Fig 4, with a correlation coefficient of 0.853, which is a very strong correlation. Comparing with similar studies [41, 42], this identification accuracy is acceptable considering errors such as the number of mobile subscribers

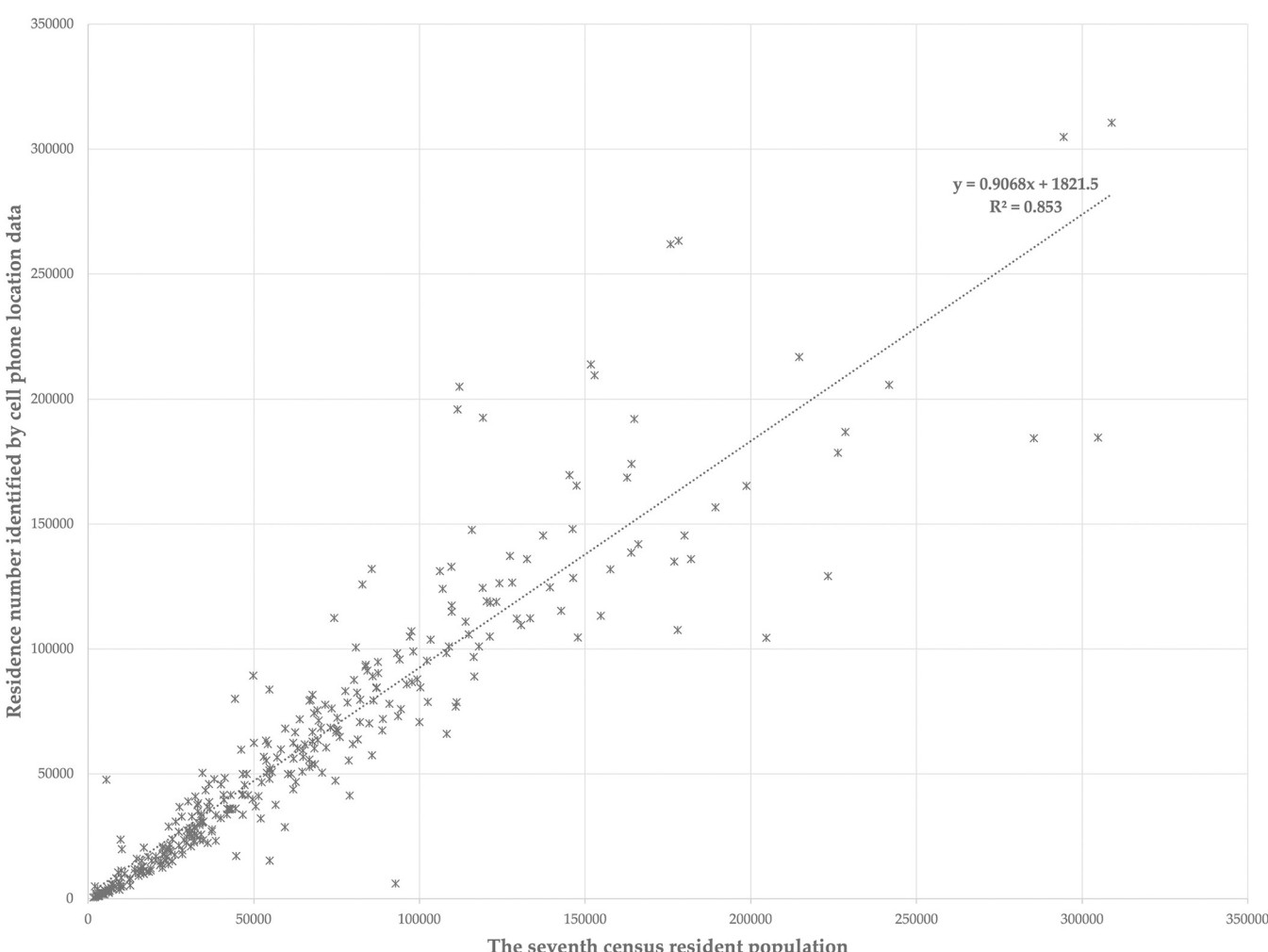

**Fig 4. Check with the resident population data of the seventh population census.**

and the inconsistency in the percentage of employed people who fit the employment pattern of this paper across streets. The places of residence identified by this method can basically reflect the real spatial distribution of residence. The test on the accuracy rate of workplace identification is not possible at this stage because the original data of the economic census is not publicly available. Considering that this paper only studies employed people who stay in fixed workplaces for a certain period using the DBSCAN algorithm, and their spatio-temporal trajectory characteristics in workplaces are more like those in residence, the workplaces identified by the residence identification method should also basically reflect the real spatial distribution of workplaces of employed people.

## 4.2 Employment center identification

This research uses the above identified data on the workplace and place of employment and commuting time of the employed population in Beijing in May 2021 to generate an employment density contour distribution map using data aggregated to a 100 m * 100 m raster grid, combined with kriging spatial interpolation, as shown in Fig 5.

In this paper, using cell phone location, path planning, user portrait and behavior data in Beijing in May 2021, we use the spatial clustering DBSCAN algorithm to cluster the location

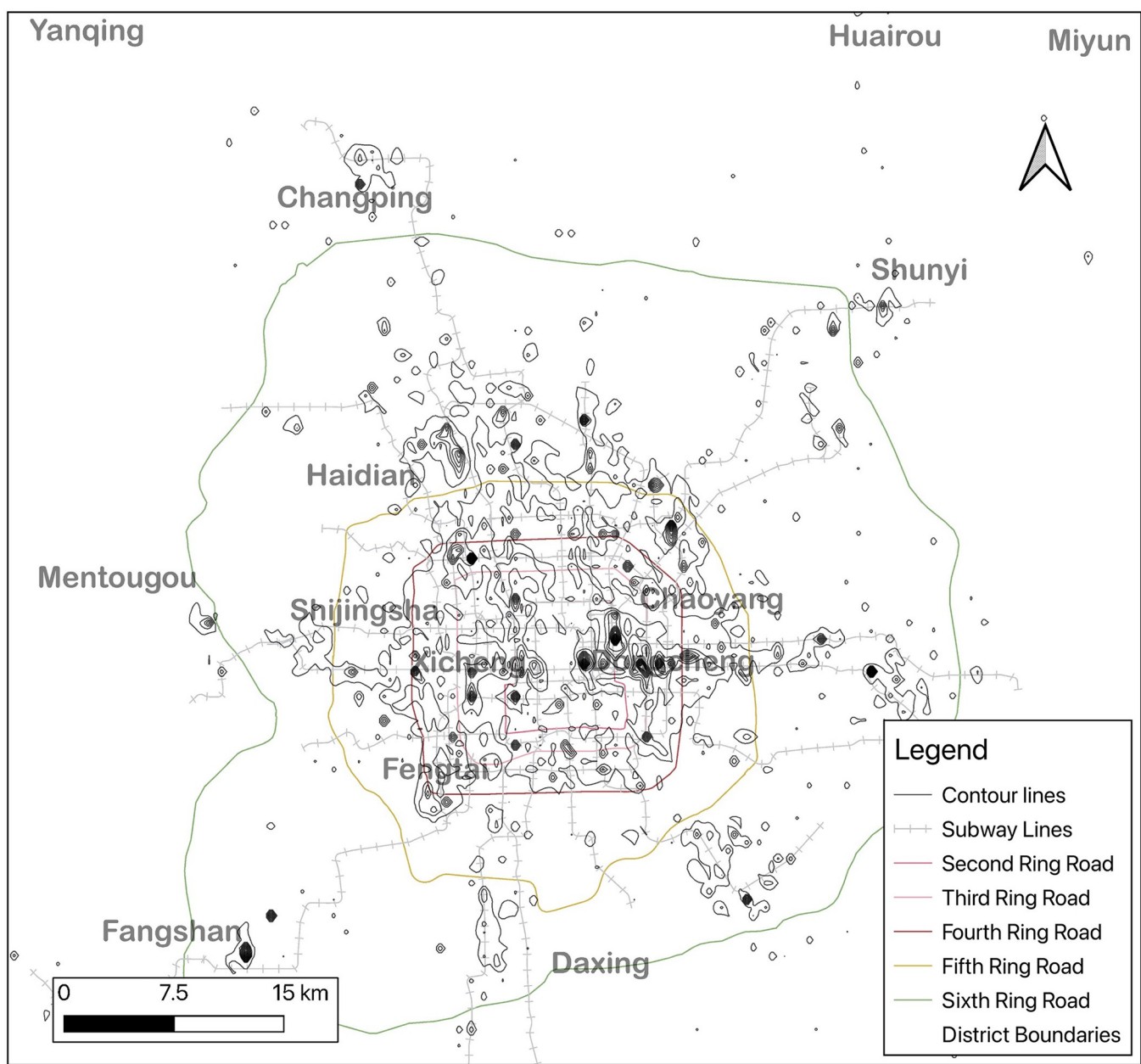

**Fig 5. Contour lines of employment place.** Reprinted from [https://www.resdc.cn/, accessed on 10 May 2021], original copyright 2021.

points and get multiple independent user clusters, extract features such as location, land use attributes and user portrait from the user clusters, classify the user clusters using machine learning algorithms such as XGBoost, identify the employed population in Beijing in May 2021 of residence and employment, and spatially cluster the employed population on a 100 m * 100 m raster grid, and use kriging spatial interpolation to generate equipotential surfaces, as shown in Fig 3, and combine local empirical knowledge to define the employment center as an equipotential surface enclosed area with an area of 1 to 10 square kilometers, a population density greater than 10,000 persons per square kilometer, and a number of people in the area of no less than 20,000. This is used to identify the employment centers in the mega-metropolitan area of Beijing.

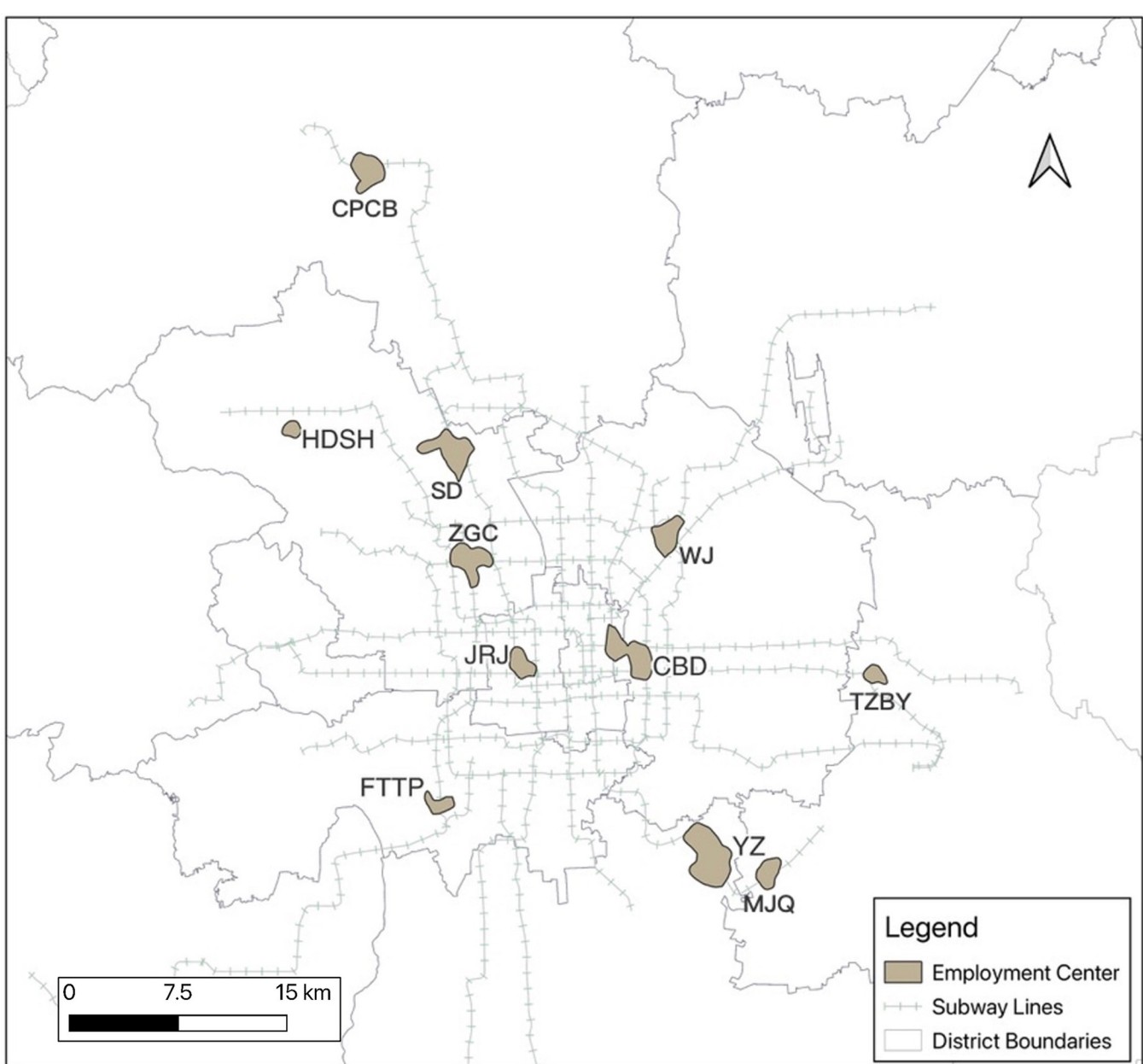

**Fig 6. Employment center.** Reprinted from [https://www.resdc.cn/, accessed on 10 May 2021], original copyright 2021.

Based on local empirical knowledge, employment centers are defined as areas enclosed by equipotential surfaces, and employment centers in the mega-metropolitan area of Beijing are identified by selecting an area of 1 to 10 square kilometers, a population density greater than 10,000 people per square kilometer, and no less than 20,000 people employed in the area, as shown in Fig 6.

Using the above methods, a total of 11 Beijing employment centers were identified, including Shangdi(SD), Wangjing(WJ), Zhongguancun(ZGC), Jinrongjie(JRJ), CBD, Yizhuang Economic Development Zone(YZ), Changping Chengbei(CPCB), Haidian Shanhou(HDSH), Tongzhou Beiyuan(TZBY), Fengtai Technology Park(FTTP) and Majuqiao(MJQ). From the

perspective of spatial distribution, the employment centers show a significant multi-center distribution, among which, in the functional core areas of the capital (the eastern and western cities), the financial street shows a high concentration of employment; In the four districts of the central city (Chaoyang, Haidian, Fengtai and Shijingshan), the CBD and Zhongguancun in the middle of the city show a high concentration of employment. The Fengtai Technology Park in the southwest, Wangjing in the northeast, Shangdi in the northwest, and Haidian Shanhou in the rear of the region play a sustained impact, and the driving role in the surrounding areas is highlighted to form the employment center; The employment centers identified in the suburbs mainly include Tongzhou Beiyuan, Changping Chengbei, and the development of Yizhuang Economic Development Zone and Majuqiao. In the suburbs, the employment centers are not identified because of the unclear employment gathering situation.

The area of 11 employment centers in Beijing is 43.21 square kilometers, of which 1.02 million people are employed, accounting for 7.5% of the total employment of the city. There are 7 employment centers located in the central city, accounting for 79.7% of the total employment centers, while the number of jobs accounts for nearly 79.8% of the main employment centers, which is equivalent to the proportion of the area, indicating that the overall employment density of the employment centers in the central city is not significantly higher than that of the suburban employment centers. From the perspective of the employment density of a single employment center, the employment density of CBD, Financial Street, Zhongguancun and Fengtai Science and Technology Park is relatively high. The number of jobs per unit area from the employment-intensive area is more than 30,000, while the employment density of Yizhuang Economic Development Zone and Changping Chengbei is relatively low. From the perspective of total employment, the employment scale of CBD is considerable, accounting for 21.6% of the employment scale of the whole employment center; The employment of Shangdi and Zhongguancun is characterized by high density and large area. Not only is the employment density high, but also the total employment is relatively large (Table 3).

Analyze the causes of the spatial distribution of these high-density employment centers, the path dependence of early urban planning, the recent industrial promotion and evolution, and land transfer policies may be important factors affecting the spatial distribution of high-density employment centers in the main urban area. As early as the early days of the founding of the People's Republic of China, in the first draft of Beijing's urban planning, the Jianguomen-Dawanglu area in Chaoyang District was divided into industrial construction land and became one of the areas with highly concentrated employment. After experiencing the turmoil of the Cultural Revolution and the recovery growth period at the beginning of the reform and

**Table 3. Spatial characteristics of employment centers in Beijing.**

| Employment Center | Area (square kilometer) | Employment | Employment Density | Jobs as a proportion of total employment centers |
|---|---|---|---|---|
| JRJ | 2.596375736 | 99971 | 38504.05726 | 9.8% |
| FTTP | 1.783135859 | 58196 | 32636.88502 | 5.7% |
| TZBY | 1.49002125 | 22444 | 15062.87243 | 2.2% |
| HDSH | 1.162817846 | 22810 | 19616.14201 | 2.2% |
| SD | 5.845251743 | 166952 | 28561.98627 | 16.4% |
| ZGC | 4.762491911 | 155965 | 32748.61205 | 15.3% |
| YZ | 9.304873251 | 100579 | 10809.28211 | 9.9% |
| CPCB | 4.227182517 | 47318 | 11193.74425 | 4.6% |
| WJ | 3.545414893 | 90224 | 25448.07948 | 8.8% |
| MJQ | 2.526010648 | 36346 | 14388.69627 | 3.6% |
| CBD | 5.964712664 | 220084 | 36897.67008 | 21.6% |

opening up, Beijing first proposed the idea of building an urban CBD in 1993, and a large number of foreign-related office buildings and hotels rose here. The Jianguomen and Dawanglu areas have not only become the earliest opening windows of Beijing, but also become one of the most popular and headquarters economy gathering areas. With the establishment of the "Beijing New Technology Industrial Development Pilot Zone" in 1988, the first national high-tech industrial development zone was established here and became increasingly prosperous. With the strong scientific and technological support of famous universities and research institutions at home and abroad, it has become the most densely populated area of high-intelligence talents in Beijing. In addition, the financial street in the Guang'anmen area of Fuwai, the transportation hub employment area near Taipingqiao, and Jiuxianqiao Electronic Industrial Park all have the role of the government in guiding the planning and construction.

## 4.3 Industrial diversity and classification of employment centers

Through analyzing the data of 4.21 million industrial and commercial enterprises in Beijing, it is found that scientific research and technical services are widely distributed in Chaoyang, Haidian and Fengtai in the overall distribution of Beijing enterprises; Chaoyang, Xicheng and Haidian have obvious advantages in the financial industry; The mining industry is only distributed in Fangshan and Shunyi; The manufacturing industry is relatively concentrated in Daxing, Tongzhou, Shunyi, Changping and other regions; The production and supply of electric water are mainly concentrated in Changping, Fangshan, Tongzhou and other regions; The distribution of construction industry is relatively balanced, but the proportion of Chaoyang and Fengtai is large; Wholesale and retail businesses are widely distributed in all regions, but Fengtai, Chaoyang and Haidian have comparative advantages; There are a large number of transportation and warehousing units in Chaoyang, Daxing, Fengtai and other places; Chaoyang and Haidian have obvious advantages in accommodation and catering industry; Haidian's information technology industry has absolute advantages, and Chaoyang and Changping also have certain distribution; Chaoyang and Haidian have a high share of the real estate industry; Leasing and business service industry, water conservancy environment and public facilities management industry, residential service industry, education industry, health industry, culture, sports and entertainment industry are similar in distribution, Chaoyang and Haidian are more distributed, followed by Dongcheng and Xicheng; Public management and social security organizations are equally distributed in Xicheng, Dongcheng, Chaoyang, Haidian and other areas, and also in Fangshan and Daxing.

This paper uses the Shannon Wiener Diversity Index to calculate the industrial diversity of 11 employment centers based on the industrial and commercial registration data of enterprises. The study found that the top three types of enterprises with registered capital in the 11 employment centers are commercial services, monetary financial services, science and technology promotion and application services. The most registered enterprises in the 11 employment centers are Zhongguancun, 30772, and the least registered enterprises are Wangjing, There are only 1261 registered enterprises. On the whole, the types of registered enterprises are mainly business services, monetary and financial services, science and technology promotion and application services. Select the top 10 enterprises with the largest registered capital to make radar charts, as shown in Fig 7.

The number of registered enterprises, the total amount of registered capital and the spatial distribution of the employment centers can reflect the degree of concentration and difference of each industry in space. By observing the number of enterprises and the scale of registered capital in each industry of each employment center, and analyzing the advantages of their distribution, it is found that the total registered capital of enterprises in the business service and

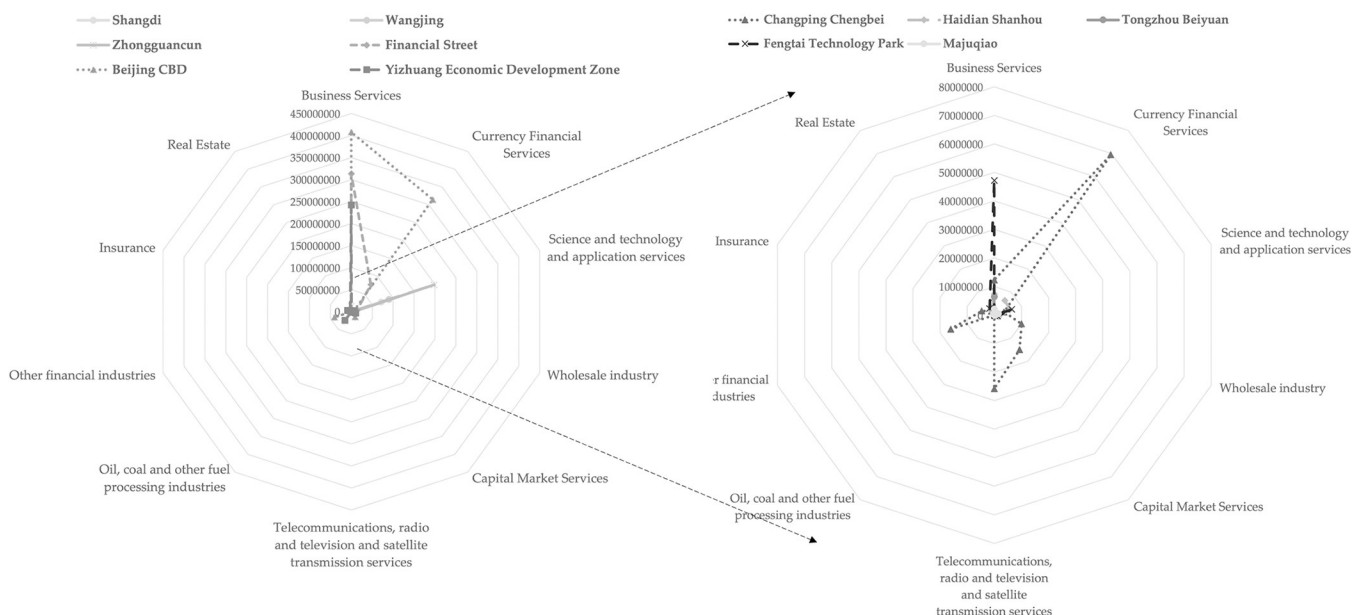

**Fig 7. Radar map of total registered capital of different industry types(TOP10) in the employment centers.**

currency financial service industries in CBD is the highest, while the financial street and Yizhuang Economic Development Zone are mainly in the business service industry, Zhongguancun The type of enterprises with the highest total registered capital in Wangjing and Shangdi is the technology promotion and application service industry, while the total registered capital advantage of other employment centers is not obvious.

This paper uses the Shannon Wiener Diversity Index, which describes biodiversity, to analyze the industrial diversity of the employment center. It mainly uses the types of enterprises in the employment center and the registered capital to calculate the industrial diversity of the employment center. It is mainly used to measure the species richness within the system and the distribution uniformity of individual enterprises in various categories, The average commuting time of employees in 11 employment centers in Beijing is calculated using mobile phone positioning data, and the industrial diversity and average commuting time of 11 employment centers in Beijing are finally obtained as shown in Table 4.

This paper uses the industrial diversity index of the employment center and the average commuting time of the employees as the basis for the classification of the employment centers, combines the characteristics of the employment centers, uses the K-Means classification algorithm and selects the classification number K = 3 according to experience to classify the identified Beijing employment centers, and the classification results are shown in Fig 8.

According to the analysis of Table 4 and Fig 8, 11 employment centers in Beijing are divided into the following three categories: (1) CBD, Zhongguancun, Wangjing and Fengtai Science and Technology Park; (2) Financial Street, Shangdi, Yizhuang Economic Development Zone, Majuqiao and Haidian Shanhou; (3) Changping Chengbei and Tongzhou Beiyuan. It

**Table 4. The attributes of residential employment and commuting time.**

| Type/EmployeeCenter | CPCB | HDSH | SD | WJ | ZGC | JRJ | CBD | TZBY | FTTP | MJQ | YZ |
|---|---|---|---|---|---|---|---|---|---|---|---|
| Industry Diversity | 0.416 | 0.594 | 0.498 | 0.380 | 0.383 | 0.430 | 0.395 | 0.434 | 0.320 | 0.560 | 0.307 |
| Commuting time (hours) | 0.586 | 0.796 | 0.776 | 0.893 | 0.866 | 0.727 | 0.846 | 0.619 | 0.848 | 0.747 | 0.750 |

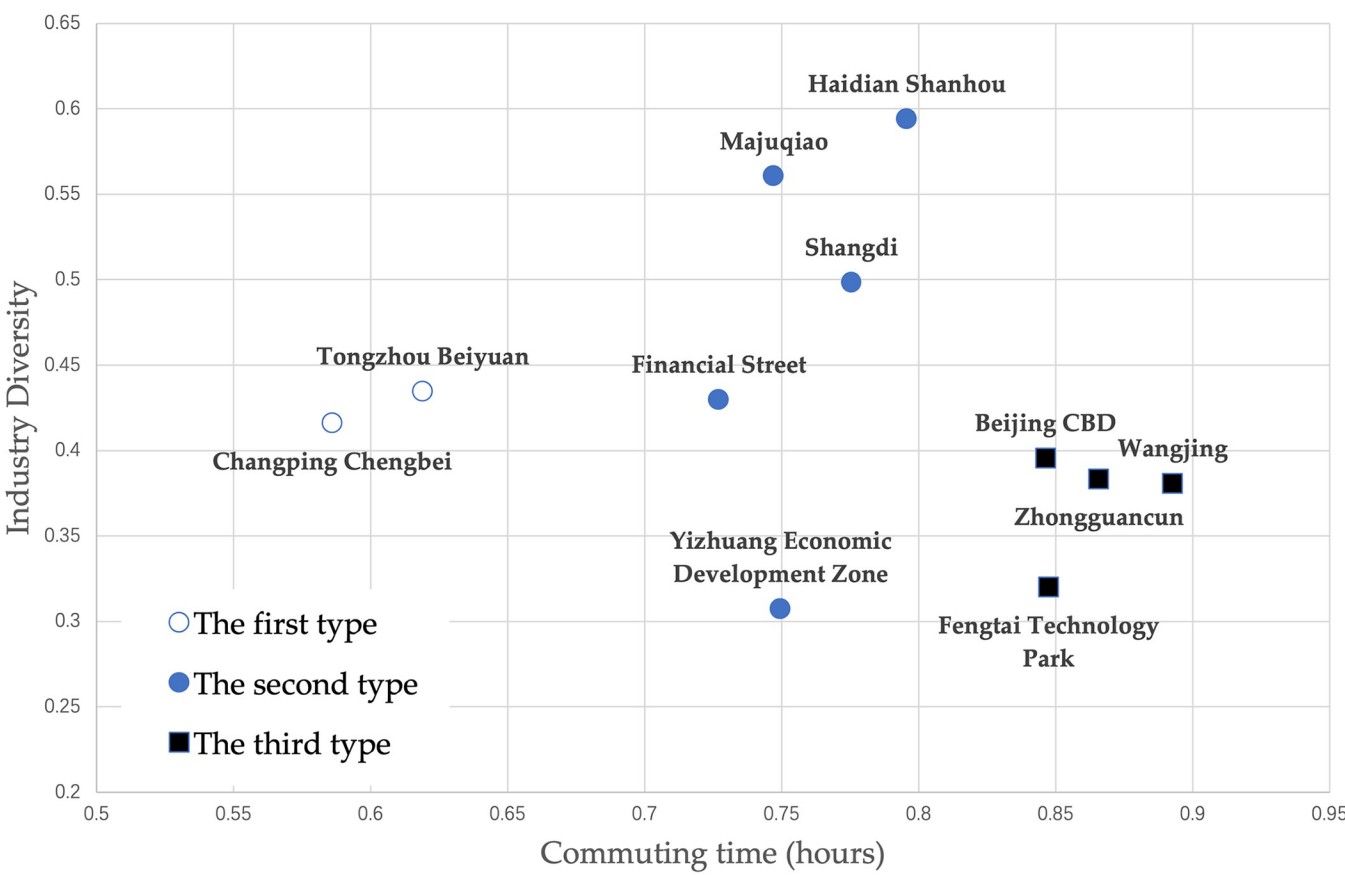

**Fig 8. Employment center categories.**

can be seen from the classification results that the first type of employment centers CBD, Zhongguancun, Wangjing and Fengtai Science and Technology Park have become important employment centers in Beijing. Their overall advantage industries are prominent. Because they are relatively close to the city center, and the number of employees is large, their employees need to come to work from a farther region, resulting in a longer average commuting time of the employees. Among the second type of employment centers, Financial Street is relatively reasonable in terms of the size of the employment center and the industrial diversity index. Analyzing the space of its employees, it is found that its unique transportation accessibility and the relatively high income of the financial industry enable the employees to afford the relatively high housing costs in the city center, while Shangdi, Majuqiao and Haidian Shanhou are relatively large in terms of the industrial diversity index, mainly because they are relatively far from the city center in space, It needs a relatively complete industrial layout to be self-sufficient. In Yizhuang Economic Development Zone, the smallest industrial diversity index also reflects its focus on advantageous industries and drive the development of employment centers. The third type of employment centers, Changping Chengbei and Tongzhou Beiyuan, are located in the suburbs relatively far from the city center, and have corresponding residential space around them, which makes the average commuting time of their employees shorter.

Here we try to analyze the driving force of the emergence of multi-center in Beijing. On the surface, the government's planning and policy guidance have played a great role. CBD, Financial Street, Zhongguancun, Shangdi and so on are the key development areas determined by the relevant planning. Under the conditions of regional economy, the location choice of

enterprises is completely based on their own interests, and the most important is the pursuit of agglomeration benefits. The reason why these regions have grown into employment centers may be that the relatively relaxed control of land use nature and development intensity in the planning has given opportunities and space for agglomeration economic growth. Of course, this judgment still needs to be demonstrated with more planning data in the future.

## 5. Discussion

Based on the above research results, the employment population in Beijing has obvious agglomeration characteristics in the central urban area. This paper identifies 11 employment centers in Beijing, and CBD is still the largest employment center. Zhongguancun and Shangdi are close to the employment density and scale of CBD. Although both are located in the central urban area, they are still a certain distance from the city center. For many years, Beijing has tried to reduce the pressure on the central city by restricting the scale of the central city, developing rail transit, and improving infrastructure and other measures to guide the city to develop into a multi-center pattern, and has achieved initial results. Zhongguancun has rapidly developed into one of the major scientific and technological employment centers with the same scale as CBD.

Compared with previous studies, the use of mobile phone location data and micro-enterprise data in this paper makes the spatial expression of employment density more continuous, and the identification of employment center boundaries clearer. With regard to the measurement method of employment center system, mobile phone signaling data is used, and the space unit identified by the employment center breaks away from the restriction of the established space unit (generally street), and is Under the 100 m × 100 m grid, it can basically be identified according to the actual scope of the employment center, which is helpful to identify the actual employment centers that are difficult to identify according to the average employment density of the space unit due to the large size of the space unit they belong to, and also help to identify the employment centers that are difficult to delimit the real boundary due to crossing the space unit. At the same time, compared with the permanent population, the employed population is a more dynamic factor. Due to the differences in the factors affecting the employment location and residence location, the separation of employment and residence of the employed population is the main reason for the formation of urban commuting pressure. This paper clearly identifies the boundaries of the hot areas where the employed population congregate and the number and space of the employed population, as well as the public transportation arrangement Public service arrangements such as public security and emergency response have important reference significance.

Although there may be errors of tens of meters in mobile positioning, the spatial resolution of mobile positioning data is still acceptable when measuring employment centers in Beijing. The economic census has the real employment data of the employed, and the population census has the real residence data of the employed, which is more accurate than the employment and residence data identified by mobile phone signaling data. However, there is no link between the place of employment in the economic census and the place of residence in the population census. Compared with traditional data, mobile positioning data is characterized by the ability to obtain the employment and residence of the employees at the same time, and establish the spatial connection between the two and the time required for commuting. However, according to the current identification method, it is impossible to identify the work place and residence place without fixed work place (such as taxi driver) at the same time. How to improve the recognition rate of these users needs further research. In addition, the mobile phone location data does not involve social and economic attributes, and can only be used for

descriptive analysis of the employment center system, which can not accurately analyze the impact of factors such as the occupation, income, consumption and family of the employees on the employment center system. In addition, due to the inconsistency between the actual office and registration place, the registered enterprises of the enterprises in the employment center are inaccurate, but their proportion is relatively small, The impact on relatively large employment centers can be ignored.

## 6. Conclusions

To some extent, this research can help solve the limitations of previous research on the employment center system due to large spatial units and lack of commuting data in terms of center identification and commuting connection measurement, and hopes to provide help for the construction of Beijing's employment multi-center system. Based on the research on the identification and classification of job centers, we believe that the government should pay attention to the construction of job centers and formulate corresponding policies. The policies should include measures such as providing financial support and tax incentives to attract more companies to set up offices in employment centers. It also encourages high-tech enterprises to set up R&D centers and production bases in the employment centers. This will help enhance Beijing's technological innovation capability and promote industrial upgrading and transformation. In terms of transportation construction between the employment centers and the surrounding areas to improve the convenience of transportation for the employed, the infrastructure construction of the employment centers should promote the improvement of enterprise services and provide relevant consulting, training and technical support services for the employment centers to improve the innovation ability and competitiveness of the enterprises. In the new era, Beijing needs to further study the space optimization strategy and planning management policy to promote the coordinated development of employment space in order to achieve the goals and tasks of "scientifically allocate resource elements, coordinate the relationship between employment and housing, promote the balanced development of employment and housing, effectively control the 'big city disease', and build a world-class harmonious and livable city" proposed in the new version of the Beijing Urban Master Plan (2016–2035), The research can provide a reference for understanding the laws and technical analysis of employment centers, quantify the spatial structure of employment, guide the construction of a multi-center system, and adjust land use rules.

## Author Contributions

**Conceptualization:** Liang Wang.

**Data curation:** Liang Wang.

**Formal analysis:** Liang Wang.

**Software:** He Cui.

**Visualization:** He Cui.

**Writing – original draft:** Liang Wang.

**Writing – review & editing:** Liang Wang.

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
