## [Decision Letter · Decision Letter 0]

14 Mar 2023

PONE-D-23-02612Identification and Classification of Urban Employment Centers Based on Big Data: A Case Study of BeijingPLOS ONE

Dear Dr. wang,

Thank you for submitting your manuscript to PLOS ONE. After careful consideration, we feel that it has merit but does not fully meet PLOS ONE’s publication criteria as it currently stands. Therefore, we invite you to submit a revised version of the manuscript that addresses the points raised during the review process.

The reviewers have recommended some revisions to your manuscript.  Therefore, I invite you to respond to the reviewers' comments and revise your manuscript. I also agree that the authors need to add more discussions about the implications of this research on local planning. Besides, some relevant references are missing in this manuscript, such as Yu et al., Journal of Transport Geography (2015) which proposed a CBD delimitation method. It could be better to provide a wide review of the literature. 

We look forward to receiving your revised manuscript.

Kind regards,

Wenhao Yu, Ph.D.

Academic Editor

PLOS ONE

Journal Requirements:

4. We note that Figures 1,2,4 and 5 in your submission contain [map/satellite] images which may be copyrighted. All PLOS content is published under the Creative Commons Attribution License (CC BY 4.0), which means that the manuscript, images, and Supporting Information files will be freely available online, and any third party is permitted to access, download, copy, distribute, and use these materials in any way, even commercially, with proper attribution. For these reasons, we cannot publish previously copyrighted maps or satellite images created using proprietary data, such as Google software (Google Maps, Street View, and Earth). For more information, see our copyright guidelines: http://journals.plos.org/plosone/s/licenses-and-copyright.

a. You may seek permission from the original copyright holder of Figures 1,2,4 and 5 to publish the content specifically under the CC BY 4.0 license.  

Additional Editor Comments:

The reviewers have recommended some revisions to your manuscript. Therefore, I invite you to respond to the reviewers' comments and revise your manuscript. I also agree that the authors need to add more discussions about the implications of this research on local planning. Besides, some relevant references are missing in this manuscript, such as Yu et al., Journal of Transport Geography (2015) which proposed a CBD delimitation method. It could be better to provide a wide review of the literature.

Reviewers' comments:

Reviewer's Responses to Questions

**Comments to the Author**

1. Is the manuscript technically sound, and do the data support the conclusions?

Reviewer #1: Yes

Reviewer #2: Partly

2. Has the statistical analysis been performed appropriately and rigorously? 

Reviewer #1: Yes

Reviewer #2: Yes

3. Have the authors made all data underlying the findings in their manuscript fully available?

Reviewer #1: Yes

Reviewer #2: No

4. Is the manuscript presented in an intelligible fashion and written in standard English?

Reviewer #1: No

Reviewer #2: Yes

5. Review Comments to the Author

Reviewer #1: The authors have conducted a relatively complete study in which the identification and classification of Beijing urban employment centers were carried out. I think the paper is interesting but have some concerns for the authors to consider:

1. I recommend separating the study's background and current situation in the first paragraph of the introduction. And summarize the research shortcomings, leading to the research objectives.

2. The authors should summarize parts 2.1 and 2.2, respectively, summarizing the advantages and disadvantages of the research methods and data.

3. Figure 1 should have a map of China, illustrating Beijing's location in China. The legend cannot obscure the map.

4. Line 182: The reference position needs to be adjusted. Please check the entire manuscript.

5. The second paragraph of 3.2.1 is the research methodology. Please modify its position.

6. Ethics statement is not methodology. It needs to be deleted.

7. The authors had to describe the DBSCAN algorithm, XGBoost, and the K-Means clustering algorithm in detail as core methods of the study.

8. The study's results can guide Beijing's planning, so the discussion needs a separate section, especially policy recommendations.

Reviewer #2: In this manuscript, the authors proposed a new research direction on urban employment space based on mobile phone location data and micro-enterprise data: the identification and classification of employment center, so as to solve the problems of previous employment center system research due to large space units and lack of commuting data. Overall, the paper is well organized and written, but it needs some improvement before acceptance for publication.

My major comments are as follows:

In the Methodology part, four methods were used in the manuscript: Shannon-Wiener Diversity Index, XGBoost, DBSCAN, K-means. Why are those methods chosen? How to justify their appropriateness? It is suggested to add some explanations or comparisons with other similar methods to make your method more convincing. Additionally, it is recommended to draw a large flowchart to illustrate the role of these methods in different stages of the study. This will make it easier for readers to understand the authors’ intentions.

The classification of employment centers is a little confusing, it could be that I missed some important detail, I would therefore suggest the authors elaborate on the significance of the classification, or analyze the functions or influencing factors of employment centers at different levels.

The index of commuting data, the authors described it in line 258 as follows: average commuting aggregation force, but actually the authors calculated the commuting time (Table 4) and in line 396, this index became average commuting distance. There are confusion and inconsistency for this index. Please clarify its definition and make it consistent throughout the text.

In terms of the organization of the paper, firstly, it is better to have a paragraph outlining the rest of the article at the end of the introduction Section. Besides, in the conclusion part, presenting the contributions and innovations of the article in a detailed and organized manner through several points is more recommended.

Minor comments are as follows:

1. Language of the manuscript-at-hand needs to be revisited, i.e., particularly, in terms of the sentence structure and the punctuation.

2. Line 415, “The second type of employment centers” should be “The third type” to be logical.

3. Maps in figure 1, 2, 4, 5, 6 need to be supplemented with scale bar.

4. Please supplement Figure 7 with some key to the symbols to illustrate the three types of employment centers you divide.

Given the above major comments, I recommend a major revision for this manuscript before it can be considered for publication.

6. PLOS authors have the option to publish the peer review history of their article (what does this mean?). If published, this will include your full peer review and any attached files.

Reviewer #1: No

Reviewer #2: No

---

## [Author Response · Author response to Decision Letter 0]

3 Dec 2023

Response to Reviewer 1 Comments

Reviewer #1: The authors have conducted a relatively complete study in which the identification and classification of Beijing urban employment centers were carried out. I think the paper is interesting but have some concerns for the authors to consider:

Point 1: I recommend separating the study's background and current situation in the first paragraph of the introduction. And summarize the research shortcomings, leading to the research objectives.

Response 1: Thank you very much for your comment. This constructive comment is very helpful to the logic of our article. I have separated the background from the current situation of the study and summarized the shortcomings of the existing studies to draw out the research objectives.

Point 2: The authors should summarize parts 2.1 and 2.2, respectively, summarizing the advantages and disadvantages of the research methods and data.

Response 2: Thank you very much for your comments. We have summarized the advantages and disadvantages of their methods and data for 2.1 and 2.2 parts respectively. In terms of research methods, the identification of urban centers in academia is mainly based on employment density or population density data, using employment or population density peak, threshold method, parametric method and semi-parametric method to identify urban centers, but limited by the accuracy of the data, most methods identify employment center boundaries only with streets as the minimum boundary, which is not consistent with In terms of data, the traditional data acquisition methods are more laborious and lead to higher research costs.

Point 3: Figure 1 should have a map of China, illustrating Beijing's location in China. The legend cannot obscure the map.

Response 3: Thank you very much for your comments. I have added a map to illustrate the location of Beijing in China, adjusting the location of the legend on the side.

Point 4: Line 182: The reference position needs to be adjusted. Please check the entire manuscript.

Response 4: Thank you very much for your comments. I have adjusted accordingly.

Point 5: The second paragraph of 3.2.1 is the research methodology. Please modify its position.

Response 5: Thank you very much for your comments. I have adjusted accordingly.

Point 6: Ethics statement is not methodology. It needs to be deleted.

Response 6: Thank you very much for your comments. As the Ethics statement was placed in the Research Methods section in conjunction with an earlier issue of plos one as requested by the editor, I appreciate your guidance.

Point 7: The authors had to describe the DBSCAN algorithm, XGBoost, and the K-Means clustering algorithm in detail as core methods of the study.

Response 7: Thank you very much for your comments. Your constructive comments are helpful for the logic of our article, I have explained the corresponding methods, DBSCAN is a density-based clustering algorithm, the number of objects (points or other spatial objects) contained in a certain region of the clustering space is not less than a given threshold, the algorithm can discover arbitrarily shaped clusters from a collection of data with noise, XGBoost is an integrated learning algorithm, which builds a strong classifier by training multiple weak classifiers for supervised learning classification such as user's place of residence and employment, K-means is a common clustering algorithm that divides data points into K different clusters, each of which has similar characteristics, and since K-means is an unsupervised learning algorithm, it does not require prior knowledge and can be automatically based on employment center industrial diversity and the average commuting distance of employed persons to classify the employment centers.

Point 8: The study's results can guide Beijing's planning, so the discussion needs a separate section, especially policy recommendations.

Response 8: Thank you very much for your comments. I have written the conclusion and discussion sections separately and have added the planning response section.

Response to Reviewer 2 Comments

Point 1: In the Methodology part, four methods were used in the manuscript: Shannon-Wiener Diversity Index, XGBoost, DBSCAN, K-means. Why are those methods chosen? How to justify their appropriateness? It is suggested to add some explanations or comparisons with other similar methods to make your method more convincing. 

Response 1: Thank you very much for your comments. Your constructive comments are helpful to the logic of our article, and I have explained the corresponding methodology to study the innovative use of the Shannon Wiener Diversity Index, which describes biodiversity, to analyze employment center industry diversity. The Shannon Wiener Diversity Index is primarily used to measure species richness within a system and The Shannon-Wiener Diversity Index is used to measure the intra-system species richness and the homogeneity of the distribution of individuals across classes. DBSCAN is a density-based clustering algorithm in which the number of objects (points or other spatial objects) contained in a certain region of the clustering space is not less than a given threshold, and the algorithm can discover arbitrarily shaped clusters from a collection of data with noise, XGBoost is an integrated learning algorithm that builds a strong classifier by training multiple weak classifiers for supervised learning classification of users' places of residence and employment, etc. K-means is a common clustering algorithm that divides data points into K different clusters, each of which has similar characteristics, and since K means is an unsupervised learning algorithm that does not require a priori knowledge and can automatically classify employment centers based on their industrial diversity and the average commuting distance of employed people.

Point 2: It is recommended to draw a large flowchart to illustrate the role of these methods in different stages of the study. This will make it easier for readers to understand the authors’ intentions.

Response 2: Thank you very much for your comments. I have drawn flowcharts that show the data and methods used more clearly, and modified the language accordingly for good reading.

Point 3: The classification of employment centers is a little confusing, it could be that I missed some important detail, I would therefore suggest the authors elaborate on the significance of the classification, or analyze the functions or influencing factors of employment centers at different levels.

Response 3: Thank you very much for your comments, some background and summary discussions have been added to the article. For employment centers, the significance of classification is to better provide targeted planning for different types of employment centers. This paper mainly classifies employment centers in terms of their industries and the average commuting time of employment center's job seekers. On the one hand, the more homogeneous the industry of an employment center is, the less stability it is subject to, and of course, a single employment center will also produce aggregation effects (but it needs to have a dominant industry). On the other hand, the average commuting time of job seekers in employment centers affects their commuting accessibility, and from the perspective of urban planning, convenient transportation facilities need to be provided for job seekers. Therefore, different types of employment centers require different types of support and services. By classifying employment centers, these supports and services can be better located and provided.

Point 4: The index of commuting data, the authors described it in line 258 as follows: average commuting aggregation force, but actually the authors calculated the commuting time (Table 4) and in line 396, this index became average commuting distance. There are confusion and inconsistency for this index. Please clarify its definition and make it consistent throughout the text.

Response 4: Thank you very much for your comments. average commuting aggregation force is my translation error, this paper is based on the average commuting time of job seekers in job centers combined with the industrial diversity of job centers to classify job centers, the average commuting time of job seekers in job centers affects their commuting accessibility, from the perspective of urban planning, it is necessary to provide job seekers with From an urban planning perspective, it is necessary to provide convenient transportation facilities for job seekers. Thank you very much for pointing out the mistake I made, and I have revised the average commute time error in this paper.

Point 5: In terms of the organization of the paper, firstly, it is better to have a paragraph outlining the rest of the article at the end of the introduction Section. Besides, in the conclusion part, presenting the contributions and innovations of the article in a detailed and organized manner through several points is more recommended.

Response 5: Thank you very much for your suggestions and comments on the organization of the paper. The suggestions you mentioned are very practical and important to help the writer better organize and present the essay. Adding a paragraph outlining the rest of the article at the end of the introduction section can give the reader a better understanding of the structure and content of the article, and in the conclusion section, it is also very important to show the contributions and innovations of the article in a detailed and organized manner. This can help the reader to better understand the value and significance of the article.

Minor comments are as follows:

Point 1: Language of the manuscript-at-hand needs to be revisited, i.e., particularly, in terms of the sentence structure and the punctuation.

Response 1: Thank you very much for your comment. I have modified the language accordingly for a good read.

Point 2: Line 415, “The second type of employment centers” should be “The third type” to be logical.

Response 2: Thank you very much for your comment. It was my mistake and I have changed it to “The third type”.

Point 3: Maps in figure 1, 2, 4, 5, 6 need to be supplemented with scale bar.

Response 3: Thank you very much for your comment. I have added the scale bar in figure 1, 2, 4, 5, 6.

Point 4: Please supplement Figure 7 with some key to the symbols to illustrate the three types of employment centers you divide.

Response 4: Thank you very much for your comment. I have made modifications to Figure 7 and added a legend to illustrate the three types of employment centers.

---

## [Decision Letter · Decision Letter 1]

15 Feb 2024

Identification and Classification of Urban Employment Centers Based on Big Data: A Case Study of Beijing

PONE-D-23-02612R1

Dear Dr. wang,

We’re pleased to inform you that your manuscript has been judged scientifically suitable for publication and will be formally accepted for publication once it meets all outstanding technical requirements.

Kind regards,

Zahid Latif, PhD

Academic Editor

PLOS ONE

Additional Editor Comments (optional):

I hope this letter finds you well. On behalf of PLOS ONE Journal , I am delighted to inform you that your submitted research article titled "Identification and Classification of Urban Employment Centers Based on Big Data: A Case Study of Beijing" has been accepted for publication in our journal.

We would like to express our sincere appreciation for your significant contribution to the field of enterprise development. Your research findings and insights are valuable additions to the scholarly discourse in our community, and we believe that your work will make a meaningful impact on the academic community and beyond.

The rigorous review process highlighted the importance and quality of your research, and we commend your dedication and expertise in conducting this study. We anticipate that your article will stimulate further discussion, inspire future research endeavors, and contribute to advancing knowledge in your area of expertise. We are honored to have the opportunity to feature your work in our publication.

Reviewers' comments:

Reviewer's Responses to Questions

**Comments to the Author**

1. If the authors have adequately addressed your comments raised in a previous round of review and you feel that this manuscript is now acceptable for publication, you may indicate that here to bypass the “Comments to the Author” section, enter your conflict of interest statement in the “Confidential to Editor” section, and submit your "Accept" recommendation.

Reviewer #3: All comments have been addressed

2. Is the manuscript technically sound, and do the data support the conclusions?

Reviewer #3: Yes

3. Has the statistical analysis been performed appropriately and rigorously? 

Reviewer #3: (No Response)

4. Have the authors made all data underlying the findings in their manuscript fully available?

Reviewer #3: Yes

5. Is the manuscript presented in an intelligible fashion and written in standard English?

Reviewer #3: Yes

6. Review Comments to the Author

Reviewer #3: COMMENTS:

I read the manuscript titled “Identification and Classification of Urban Employment Centers Based on Big Data: A Case Study of Beijing” with keen interest; the idea is worth investigating and well fits to the scope of PLOS ONE. I am thankful to the authors for submitting an interesting manuscript to the PLOS ONE Journal. However, the manuscript has some issue that needs to be corrected before final publication. The study should be enriched with the following related studies regarding Big Data Analytics & Economic Growth: The list of missing research includes (but is not restricted to):

• Big data challenges: Prioritizing by decision-making process using Analytic Network Process technique. DOI: 10.1007/s11042-017-5161-4.

• A Review of Policies concerning development of Big Data Industry in Pakistan. DOI: 10.1109/ICOMET.2018.8346315.

• A New Solution for City Distribution to Achieve Environmental Benefits within the Trend of Green Logistics: A Case Study in China. DOI: 10.3390/su12208312.

• OVERALL EVALUATION

I. Accept

II. Overall, the manuscript deserves to be published in “PLOS ONE”. I would suggest the author consider the above comments and revise the manuscript accordingly. Lastly, although a potential contribution to the research knowledge, I would suggest checking the word limit with the scope of the journal.

7. PLOS authors have the option to publish the peer review history of their article (what does this mean?). If published, this will include your full peer review and any attached files.

Reviewer #3: No

---

## [Editor Report · Acceptance letter]

26 Mar 2024

PONE-D-23-02612R1 

PLOS ONE

Dear Dr. wang, 

I'm pleased to inform you that your manuscript has been deemed suitable for publication in PLOS ONE. Congratulations! Your manuscript is now being handed over to our production team.

Kind regards, 

on behalf of

Dr. Zahid Latif 

Academic Editor

PLOS ONE